# Long-Term Results of Kidney Transplantation in Patients Aged 60 Years and Older

**DOI:** 10.3390/jcm14010078

**Published:** 2024-12-27

**Authors:** Jacek Ziaja, Damian Skrabaka, Aleksander J. Owczarek, Monika Widera, Robert Król, Aureliusz Kolonko, Andrzej Więcek

**Affiliations:** 1Department of General, Vascular and Transplant Surgery, Medical University of Silesia, 40-027 Katowice, Poland; monika.widera@sum.edu.pl (M.W.);; 2Department of Vascular and General Surgery, Provincial Specialist Hospital, 41-902 Bytom, Poland; skrabakadamian@gmail.com; 3Health Promotion and Obesity Management Unit, Department of Pathophysiology, Medical University of Silesia, 40-752 Katowice, Poland; aowczarek@sum.edu.pl; 4Department of Nephrology, Transplantation and Internal Medicine, Medical University of Silesia, 40-027 Katowice, Poland; uryniusz@wp.pl (A.K.);

**Keywords:** end-stage renal disease, kidney transplantation, elderly recipients, extended criteria donor, pair analysis

## Abstract

**Background/Objectives:** The results of kidney transplantation (KTx) in elderly patients are deteriorated by more frequent use of organs procured from older or extended criteria donors (ECDs). To eliminate the influence of donor factors on the transplantation results, the pair analysis method was applied. The study aimed to assess the survival, during long-term follow-up after transplantation, of recipients and transplanted kidneys, graft function, and factors influencing survival in recipients aged 60 years and older (≥60) compared to recipients aged less than 60 years (<60) who received a kidney from the same brain death donor (DBD). **Methods:** The study group consisted of 213 consecutive patients ≥60 who received a kidney procured from a DBD from whom the second procured kidney was transplanted to a patient <60 (control group). **Results:** The survival rates in the 10-year follow-up period were lower in those recipients ≥60 than in the control group (*p* < 0.01). The survival rates of the transplanted kidneys were comparable in both groups, while the death-censored graft survival was higher in those patients ≥60 than in the control group (*p* < 0.05). The estimated glomerular filtration rate levels were similar in the follow-up in patients ≥60 compared to those recipients <60. Transplantation of kidneys procured from ECD, longer duration of dialysis treatment, and longer duration of cold ischemia time (CIT) affected the recipient and graft survival in the ≥60 patients. Death-censored kidney graft survival was influenced by transplantation of a kidney procured from ECD, longer CIT, and the need for reoperation in the early postoperative period. **Conclusions**: Despite the higher mortality in the kidney recipients ≥60, the long-term results of KTx after the elimination of donor-dependent factors are similar or better compared to the recipients <60. The factors influencing the results of KTx in elderly recipients differ from those observed in the entire cohort and younger patients.

## 1. Introduction

The aging of the population is one of the most important factors affecting contemporary transplant medicine.

In 1990, persons over 65 years of age accounted for 10.2% of the Polish population, and, in 2022, they accounted for 19.4% [1]. It is predicted that, in 2050, persons older than 65 years will constitute 32.7% of the country’s population [2,3]. As a consequence, the number of elderly patients with end-stage renal disease (ESRD) treated with dialysis and subjected to kidney transplantation (KTx) is increasing. Among the patients starting dialysis in Poland in 2020 and among all the patients with ESRD treated with renal replacement methods, 65% and 56% were over 65 years of age, respectively [4]. In 2020, among the patients with ESRD who underwent KTx in Europe, 36% were elderly [5].

One of the most important factors that have a significant impact on the outcome of KTx includes cardiovascular diseases (CVDs). According to data from US registries, CVDs were the most common causes of death in kidney graft recipients in the one-year follow-up period (24.7%) and in the 10-year follow-up period (14.6%). In addition, cardiovascular events remain a leading cause of death in patients with good kidney graft function [6,7].

The incidence of diabetes mellitus (DM) and arterial hypertension (AH), which are the main factors contributing to the faster progression of CVDs, is increasing significantly in elderly patients [8,9]. In Poland, 13.5% of citizens aged 60 and over suffer from coronary artery disease (CAD), 63.3% from AH, and 21.5% from DM [10]. Elderly people are also at higher risk of multimorbidity, which in the Polish population occurs in 69.3% of people aged 60–64, in 78.6% of people aged 65–69, and reaches over 84.6% in seniors aged 70–74 [10].

Due to the above factors, one can expect that the KTx results in elderly patients will differ from those obtained in younger patients, and this should be taken into account both when referring them for KTx and in postoperative care. Of note, the results of KTx in the elderly are influenced by the fact that older patients more often receive organs from older donors or extended criteria donors (ECDs) [11]. To eliminate the influence of the latter factor on the transplantation results, the pair analysis method is used, in which two age groups of KTx recipients who received a kidney from the same donor are compared [12].

The aims of the study include the following:To assess the survival of the recipients and transplanted kidneys, the function of the transplanted kidneys, and the frequency of complications;To analyze the factors influencing the survival of the recipients and transplanted kidneys in recipients aged 60 years and older compared to those recipients aged less than 60 years who received a kidney from the same deceased donor during long-term follow-up after transplantation.

## 2. Materials and Methods

The analysis was carried out on a registry of all patients undergoing kidney transplantation at the transplant center in Katowice, Poland, in which information on patients operated on was gathered at baseline and also during the long-term follow-up.

Since in Polish organ allocation system patients aged 60 and older are considered as older potential recipients [13], we enrolled 213 consecutive patients in the study group with ESRD aged 60 years or older who received a kidney procured from a brain death donor (DBD) from whom the second procured kidney was transplanted to a patient under 60 years of age in the observation time between 1998 and 2020. The mentioned group of 213 kidney recipients under 60 years of age constitute the control group.

The minimal follow-up period for each patient was 2 years unless the patient died or the graft was lost earlier. The characteristics of the donors are presented in Table 1.

The recipient’s characteristics, including clinical parameters and factors related to the transplantation procedure, with their comparison between the study group and the control group, are presented in Table 2.

Early surgical complications requiring surgical intervention were venous or arterial thrombosis, hematoma or hemorrhage, urine leakage, and ureteral stenosis.

The following parameters were compared during the average follow-up time of 5.8 ± 3.1 years (from 0 to 24 years):Survival of recipients and transplanted kidneys, including death-censored graft survival;The function of transplanted kidneys based on estimated glomerular filtration rate value (eGFR) calculated according to the MDRD (Modification of Diet in Renal Disease) short formula;Occurrence of complications that may influence long-term results:(a)acute rejection episodes (AR), cytomegalovirus (CMV) infection, post-transplant DM, and cancer;(b)CVDs and their complications, or the need for interventional management related to CAD: myocardial infarction (MI), percutaneous coronary angioplasty (PTCA), coronary artery bypass grafting (CABG), stroke, and transient ischemic attack (TIA).

Additionally, the analysis of the impact of factors dependent on the donor, recipient, and transplantation procedure, as well as complications during the period after transplantation, on the survival of recipients and transplanted kidneys (including death-censored graft survival) was also performed.

### Statistical Analysis

Statistical analysis was performed using STATISTICA 13.0 PL (Tibco Software Inc., Palo Albo, CA, USA) and R software (v 4.4.0; R Development Core Team (2008). R: A language and environment for statistical computing. R Foundation for Statistical Computing, Vienna, Austria). Statistical significance was set at a *p*-value below 0.05. All tests were two-tailed. Imputations were not conducted for missing data. Nominal and ordinal data were expressed as percentages, while interval data were expressed as mean value ± standard deviation in the case of a normal distribution or as the median and lower and upper quartile in the case of data with skewed or non-normal distribution. The distribution of variables was evaluated by the Shapiro–Wilk test, the quantile–quantile graph, and the Cullen–Frey graph. The homogeneity of variances was assessed by the Levene test. For comparison of interval data, the Student’s *t*-test for independent samples or the Mood’s median test was used depending on the data distribution. Categorical variables were compared using χ^2^ or Fisher exact tests. Kaplan–Meier estimates were used to assess the patients’ as well as kidney graft survival (including death-censored) according to groups of recipients aged 60 years and older (≥60) and recipients aged less than 60 years (<60). Comparisons between curves were conducted with the log-rank test. The analysis of creatinine levels through time in both groups was completed with the mixed models for repeated measures (package ‘mmrm’). The survival analysis was conducted with Cox proportional univariate and multivariate regression. The proportionality assumption was tested based on the Schoenfeld residuals (function ‘cox.zph’). Multiple collinearity was checked based on the correlation matrix of coefficients of the survival model.

## 3. Results

In our study, a significant amount of kidneys were procured from donors with extended criteria (Table 1).

Those recipients aged 60 years and older had more than twice higher incidence of DM (OR = 2.15; 95% CI: 1.25–3.71) and almost four times higher incidence of CVDs (OR = 3.81, 95% CI: 2.24–6.47), as well as slightly higher BMI compared to the younger recipients who were observed (Table 2). They were also less frequently re-transplanted, and Simulect was more often used in the induction therapy.

The survival rates in the first year after transplantation were comparable in the analyzed groups, but, in the 10-year follow-up period, the survival was lower in the group of recipients aged 60 years and older than in the control group (Table 3; Figure 1).

The survival rates of the transplanted kidneys during the entire follow-up period were comparable in both groups (Figure 2), while the death-censored graft survival was higher in the elderly recipient group than in the younger recipient one (Figure 3).

The estimated glomerular filtration (eGFR) levels up to 10 years after transplantation were similar through the follow-up in the patients aged 60 years and older compared to the control group (Table 4). For those recipients aged 60 years and older and for the control group, there were significant changes (deterioration) in eGFR through time (*p* < 0.01 and *p* < 0.001, separately).

No differences between the groups were observed in the incidence of episodes of acute rejection and CMV infection during the post-transplant period (Table 5). A trend was observed as to the higher incidence of new-onset DM, and, consequently, a higher incidence of this disease was found in the patients aged 60 years and older (OR = 2.15, 95% CI: 1.42–3.26). More than double the incidence of cardiovascular diseases, their complications, or the need for related interventions (OR = 2.09, 95% CI: 1.17–3.74) and almost double the incidence of cancer (OR = 1.88, 95% CI: 1.01–3.52) were also observed in the elderly recipients as compared to the younger ones.

The analysis of both univariate and multivariate regression, in addition to the effects of some factors on the survival of the recipients, kidney grafts, and death-censored grafts, showed significant differences depending on whether the group of all the patients or individual age groups of recipients were analyzed. In the univariate analysis, the survival of the recipients aged 60 years and older was negatively affected by transplantation of a kidney taken from an ECD, a longer duration of dialysis treatment, and a longer time of cold renal ischemia, as well as the use of ATG induction therapy and kidney re-transplantation (Appendix A). In addition to the five parameters mentioned above, the age of the donor, HLA II incompatibility, and the need for reoperation of the recipient due to surgical complications in the early period after transplantation also had an impact on the survival of the transplanted kidneys in the recipients aged 60 years (Appendix A). From the five parameters affecting recipient survival, the need for recipient reoperation in the early post-transplant period was affecting death-censored kidney survival instead of CIT duration in the recipients aged 60 years and older (Appendix A).

The multivariate analysis confirmed the effect of the transplantation of kidneys procured from ECDs, longer duration of dialysis treatment, and longer duration of CIT on the survival of recipients aged 60 years and older and kidney transplants in this group of patients (Table 6 and Table 7). As regards the impact of individual factors on death-censored kidney graft survival, in addition to kidney transplantation from ECDs and a longer CIT, the need for reoperation in the early period after transplantation influenced this survival in the long-term follow-up period (Table 8).

## 4. Discussion

The most important finding of our study is that the factors influencing the results of KTx in elderly recipients in the long-term follow-up period differ from those observed in younger patients. Additionally, we have documented that the transplantation of a kidney procured from an extended criteria donor worsens the long-term results of KTx the most in elderly recipients.

In our study, those recipients over 60 years of age were considered as elderly. This criterion has been adopted according to the definition of the United Nations [14,15] and has been used in a number of analyses regarding kidney transplantation [12,16,17,18,19]. Also, as previously mentioned, in the Polish organ allocation system, patients aged 60 and older are treated as older potential recipients; they have priority in choosing KTx from a donor aged over 65 years or receiving additional points in the score system [13,20]. According to other authors, older age is considered to be 65 years and older [14], and, in a number of studies, this age is referred to as an older patient. In our analysis of pairs, however, the adoption of the 65-year point would result in a significant reduction in the number of patients in the analyzed groups and would hinder statistical analysis. Interestingly, there are studies in which kidney recipients aged 60–69 years are the control group for recipients aged 70 years and older [21].

Our study was based on an ongoing registry containing detailed information on the health status of recipients over a long 10-year follow-up period [22]. It should be emphasized that, in the analysis of KTx outcomes in elderly patients, most authors present 5-year results, and, therefore, it is difficult to relate our observations to the literature data [19,23,24].

Our analysis revealed that almost 70% of the recipients were alive 10 years after transplantation. These results are slightly better than those observed by other authors for groups of recipients aged 60 years and older receiving kidneys from deceased donors, which ranged from 41–43%, as reported by Yemini et al. [25], to 55%, as reported by Sanchez-Avila et al. [26]. With the cut-off point of 65 years, the 10-year follow-up results of KTx procured from deceased donors are obviously worse, like the 45% result reported by Adani et al. [22].

Undoubtedly, the higher mortality observed in elderly recipients compared to younger recipients was associated with higher rates of patients aged 60 years and older suffering from arterial hypertension, diabetes, CVDs, and cancer; however, we were only able to demonstrate a direct effect of pre-transplant DM and CVDs on recipient survival in the stratified study group of patients and not among recipients aged 60 years and older [22,27,28]. Our results confirm that, with proper qualification of elderly patients regarding KTx, very good long-term recipient survival can be expected.

In the conducted analysis, very good results were observed in terms of survival and function regarding the transplanted kidneys in the elderly patients: 10 years after KTx, kidney function was preserved in as many as 87% of the recipients, and the eGFR level did not drop below 40 mL/min/1.73 m^2^ despite the fact that as many as 2/3 of the recipients received organs procured from ECDs. In the 5-year follow-up period, our results were comparable to those observed by Saidi et al., although in their study the percentage of ECDs was twice as low [23] and was better than that reported by Otero-Ravina et al. [19].

In our study, the kidney graft survival probability was comparable between the studied groups, but the death-censored kidney graft survival probability was higher in the older recipients. In the mentioned Otero-Ravina et al. paper, when graft survival was analyzed while excluding patients who died with a functioning graft, no significant differences were found between those patients under 60 years and the older ones [19].

Observed in our analysis difference results from the revealed direct impact of an acute rejection episodes on the survival of transplanted kidneys in recipients under the age of 60 years, which was not observed in elderly recipients. Conversely, in our study, we did not manage to confirm the increased frequency of rejection in the elderly recipients, as reported by Tullius et al. [29] and Mendonca et al. [24].

In our study, the eGFR values in the elderly recipients were comparable to those in the younger recipients during the entire period after transplantation and were only slightly lower than observed by Kandaswamy et al. in a population of adult kidney transplant recipients (from both living and deceased donors) [30]. The obtained results prove that long-term stable function of transplanted kidneys is also possible in elderly recipients.

As mentioned previously, the multivariable Cox regression analysis showed that the risk factors for death and loss of the transplanted kidney are different in elderly patients compared to younger recipients. Moreover, the factor with the greatest influence on the analyzed parameters among older recipients was the transplantation of a kidney procured from an ECD.

The issue of the effectiveness of transplanting kidneys procured from ECDs to elderly patients has been the subject of a number of studies. A recent Belgian cohort analysis revealed that, in older patients, any survival benefit with ECD transplantation versus dialysis may be small [31]. According to Schold and Meier-Kriesche, older and frailer transplant candidates benefit from accepting lower-quality organs early after ESRD, whereas younger and healthier patients benefit from receiving higher-quality organs even with longer dialysis exposure [32]. Concerning the last study, it should be emphasized that the comparison of the obtained results to the data from the United States must be undertaken with caution due to the fact that the long-term kidney graft survival rates are markedly lower in the United States compared with Europe [33].

The limitation of the study is its retrospective character based on the results from one transplant center only. However, this methodology enabled gathering a relatively large cohort of paired elderly recipients, with a long and well-documented follow-up [25], based on the registry collecting detailed information on an ongoing basis, not only about the donor, the transplantation procedure, and the perioperative period but also about the patient’s distant fate, including the function of the transplanted kidney, comorbidities, complications, and the possible death of the patient.

## 5. Conclusions

Despite the higher mortality in kidney recipients aged 60 years and older, the long-term results of kidney transplantation after the elimination of donor-dependent factors are similar or better compared to younger recipients. The factors influencing the results of kidney transplantation in elderly recipients in the long-term follow-up period differ from those observed in the entire cohort and younger patients. The parameters that affect the long-term results of kidney transplantation the most among elderly patients include the transplantation of a kidney procured from an extended criteria donor and the duration of dialysis therapy.

## Figures and Tables

**Figure 1 jcm-14-00078-f001:**
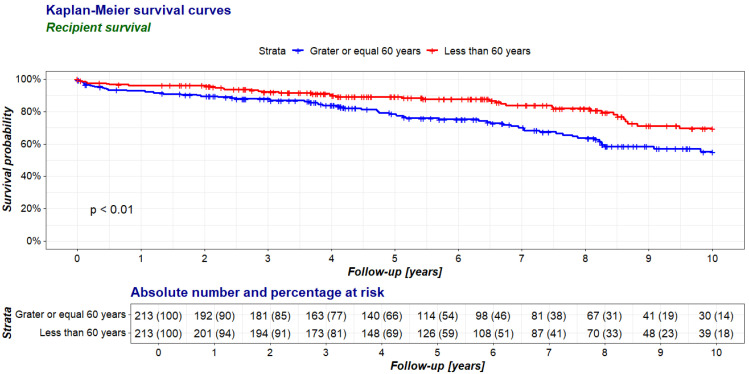
Comparison of recipient survival between recipients aged 60 years and older (≥60 group, study group) and younger than 60 years (<60 group, control group) in a 10-year follow-up period after kidney transplantation.

**Figure 2 jcm-14-00078-f002:**
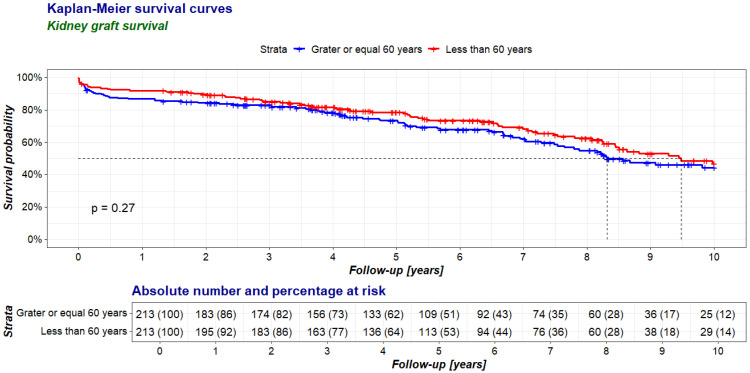
Comparison of kidney graft survival between recipients aged 60 years and older (≥60 group, study group) and younger than 60 years (<60 group, control group) in a 10-year follow-up period after kidney transplantation.

**Figure 3 jcm-14-00078-f003:**
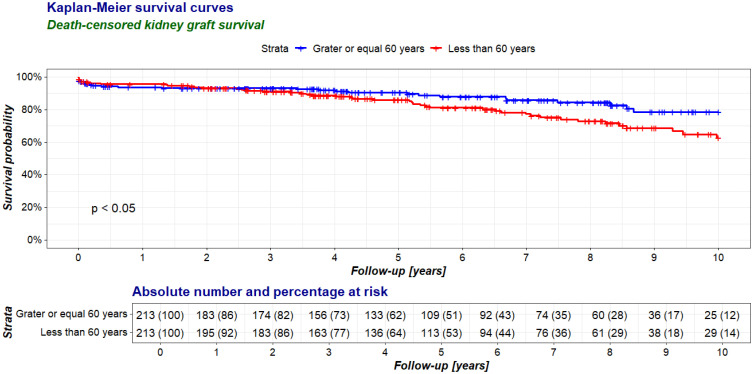
Comparison of death-censored kidney graft survival between recipients aged 60 years and older (≥60 group, study group) and younger than 60 years (<60 group, control group) in a 10-year follow-up period after kidney transplantation.

**Table 1 jcm-14-00078-t001:** The characteristics of the kidney donors: n—number; (%)—rate; mean ± standard deviation or median (upper quartile; lower quartile).

Kidney donor	n = 213
Age [years]	48.5 ± 11.1
BMI [kg/m^2^]	24.8 (23.1; 26.4)
BMI ≥ 30 kg/m^2^ [n (%)]	16 (7.6)
Gender (female) [n (%)]	80 (37.6)
Cardiovascular disease as a reason of death [n (%)]	95 (44.6)
Arterial hypertension [n (%)]	43 (20.2)
Serum creatinine [µmol/L]	97.0 (73.0; 132.0)
eGFR < 60 mL/min/1.73 m^2^ [n (%)]	43 (20.5)
Multiorgan donor [n (%)]	44 (20.7)
Extended criteria donors [n (%)]	138 (64.8)

BMI—body mass index; m—male; f—female; eGFR—estimated glomerular filtration rate.

**Table 2 jcm-14-00078-t002:** Characteristics of kidney recipients (clinical parameters and factors related to the transplantation procedure) aged 60 years and older (≥60 group, study group) and younger than 60 years (<60 group, control group): n—number; (%)—rate; mean ± standard deviation or median (upper quartile; lower quartile).

Kidney Recipient	≥60 Group	<60 Group	*p*
Recipients’ Clinical Parameters			
Age [years]	62.2 ± 3.5	44.2 ± 11.0	–
BMI [kg/m^2^]	25.8 ± 3.7	24.6 ± 3.6	**<0.01**
Gender (female) [n (%)]	81 (38.03)	85 (39.91)	0.69
Dialysis duration [months]	35.0 (22.0; 51.0)	30.5 (18.0; 53.0)	0.26
Patients with PRA > 25% [n (%)]	27 (12.7)	29 (13.6)	0.77
Re-transplantation [n (%)]	14 (6.6)	43 (20.2)	**<0.001**
Pre-transplant DM [n (%)]	44 (20.7)	23 (10.8)	**<0.01**
Arterial hypertension before KTx [n (%)]	162 (76.1)	162 (76.1)	1.00
CVD before KTx [n (%)]	65 (30.5%)	22 (10.3)	**<0.001**
Transplantation procedure factors			
Cold ischemia time [h]	17.9 ± 7.0	18.8 ± 6.6	0.18
HLA type I mismatch [points]	2.3 ± 1.0	2.3 ± 1.0	0.50
HLA type II mismatch [points]	0.6 ± 0.6	0.7 ± 0.7	0.12
Induction therapy [n (%)]	104 (48.8)	91 (42.7)	0.21
Induction therapy with Simulect [n (%)]	57 (26.8)	36 (16.9)	**<0.05**
Induction therapy with ATG [n (%)]	40 (18.8)	51 (23.9)	
No induction therapy [n (%)]	116 (54.4)	126 (59.2)	
Cyclosporine A [n (%)]	44 (20.7)	48 (22.5)	0.60
Tacrolimus [n (%)]	169 (79.3)	163 (76.5)	
Lack of mentioned above [n (%)]	0	2 (0.9)	–
Mofetetil or sodium mycofenolate [n (%)]	199 (93.4)	197 (92.5)	0.92
Azatiopryne [n (%)]	11 (5.2)	13 (6.1)	
mTOR inhibitor [n (%)]	3 (1.4)	3 (1.4)	
Double-J catheter implantation [n (%)]	34 (16.0)	27 (12.7)	0.34
Surgical complication * [n (%)]	33 (15.49)	31 (14.55)	0.79
Duration of hospitalization [days]	19.1 ± 9.8	18.8 ± 10.4	0.43

BMI—body mass index; PRA—panel-reactive antibody; DM—diabetes mellitus; KTx—kidney transplantation; CVDs—cardiovascular diseases; HLAs—human leukocyte antigens; ATG—anti-thymocyte globulin; mTOR—mammalian target of rapamycin. * Complication requiring intervention within 3 months after KTx: arterial or venous thrombosis, hematoma or bleeding, urinary leakage, or ureter stenosis.

**Table 3 jcm-14-00078-t003:** Comparison of the recipient, kidney graft, and death-censored kidney graft survival between recipients aged 60 years and older (≥60 group, study group) and younger than 60 years (<60 group, control group) in 1-, 2-, 5-, and 10-year follow-up periods after kidney transplantation.

Recipients	≥60 Group n = 213	<60 Group n = 213	*p*
1 year			
Recipient survival n(%)	198 (93.0)	205 (96.2)	0.20
Kidney graft survival n(%)	185 (86.9)	196 (92.0)	0.11
Death-censored kidney graft survival %	93.4	95.6	0.34
2 years			
Recipient survival n(%)	191 (89.7)	205 (96.2)	**<0.05**
Kidney graft survival n(%)	180 (84.5)	191 (89.7)	0.15
Death-censored kidney graft survival %	94.2	93.2	0.66
5 years			
Recipient survival n(%)	173 (81.2)	192 (90.1)	**<0.05**
Kidney graft survival n(%)	161 (75.6)	171 (80.3)	0.29
Death-censored kidney graft survival %	93.1	89.1	0.18
10 years			
Recipient survival n(%)	148 (69.5)	175 (82.2)	**<0.01**
Kidney graft survival n(%)	129 (60.6)	140 (65.7)	0.31
Death-censored kidney graft survival %	87.2	80	0.09

**Table 4 jcm-14-00078-t004:** Comparison of estimated glomerular filtration (eGFR; mL/min/1.73 m^2^) between recipients aged 60 years and older (≥60 group, study group) and younger than 60 years (<60 group, control group) in 10-year follow-up periods after kidney transplantation.

Recipients	≥60 Group			<60 Group			
Observation Period	N	Mean	SD	N	Mean	SD	*p*
3 months	171	53.8	18.3	178	56.7	20.7	0.18
1 year	167	55.6	18.5	173	56.9	21.5	0.41
2 years	162	54.3	19.1	169	55.0	21.3	0.73
3 years	152	55.0	19.2	153	55.8	22.9	0.68
4 years	126	55.6	19.9	127	54.3	21.2	0.46
5 years	98	54.8	19.0	99	53.4	22.6	0.39
6 years	84	56.3	18.3	84	50.9	21.0	**<0.05**
7 years	69	54.1	16.5	70	51.2	22.0	0.07
8 years	52	50.4	18.5	58	53.3	22.8	0.96
9 years	36	52.9	18.1	37	52.5	22.6	**<0.05**
10 years	23	48.9	17.0	28	47.6	21.1	0.12

**Table 5 jcm-14-00078-t005:** Comparison of complications after kidney transplantation between recipients aged 60 years and older (≥60 group, study group) and younger than 60 years (<60 group, control) in a 10-year follow-up period after kidney transplantation.

Recipients	≥60 Group	<60 Group	*p*
Any acute rejection n (%)	27 (12.7)	35 (16.4)	0.27
Cytomegalovirus infection n (%)	16 (7.5)	16 (7.5)	1.00
Post-transplant diabetes n (%)	42 (19.7)	28 (13.2)	0.07
Any diabetes mellitus * n (%)	86 (40.4)	51 (23.9)	**<0.001**
Cardiovascular diseases ** n (%)	38 (17.8)	20 (9.4)	**<0.05**
Cancer n (%)	30 (14.1)	17 (8.0)	**<0.05**

* Both pre- and post-transplant diabetes; ** coronary artery disease, myocardial infarction, percutaneous coronary angioplasty, coronary artery bypass grafting, stroke, or transient ischemic attack.

**Table 6 jcm-14-00078-t006:** Factors influencing recipient survival in a 10-year follow-up period after kidney transplantation—multivariate linear regression analysis. ≥60 group—recipients aged 60 years and older (study group); <60 group—recipients younger than 60 years (control group). HR—Hazard Ratio; CI—Confidence Interval; X—non-significant (removed from the model). CVD—cardiovascular disease; KTx—kidney transplantation.

	<60 Group			≥60 Group			Stratified		
Characteristic	HR	95% CI	*p*	HR	95% CI	*p*	HR	95% CI	*p*
Dialysis duration (months)	1.013	1.004, 1.022	**<0.01**	1.022	1.012, 1.031	**<0.001**	1.018	1.011, 1.024	**<0.001**
Extended criteria donors	X			1.736	1.006, 2.996	**<0.05**	X		
Cold ischemia time (h)	X			1.045	1.009, 1.082	**<0.05**	1.033	1.004, 1.062	**<0.05**
Pre-transplant diabetes	X			X			1.690	1.035, 2.762	**<0.05**
CVD before KTx	X			X			1.603	1.020, 2.520	**<0.05**

**Table 7 jcm-14-00078-t007:** Factors influencing kidney graft survival in a 10-year follow-up period after kidney transplantation—multivariate linear regression analysis. ≥60 group—recipients aged 60 years and older (study group); <60 group—recipients younger than 60 years (control group). HR—Hazard Ratio; CI—Confidence Interval; X—non-significant.

	<60 Group			≥60 Group			Stratified		
Characteristic	HR	95% CI	*p*	HR	95% CI	*p*	HR	95% CI	*p*
Multiorgan donor	0.551	0.346, 0.875	**<0.05**	X			0.698	0.505, 0.965	**<0.05**
Any acute rejection	2.671	1.600, 4.460	**<0.001**	X			1.928	1.286, 2.891	**<0.001**
Surgical complication	2.186	1.290, 3.705	**<0.01**	X			1.662	1.134, 2.435	**0.009**
Dialysis duration (mths)	1.008	1.001, 1.015	**<0.05**	1.021	1.013, 1.029	**<0.001**	1.014	1.009, 1.019	**<0.001**
Extended criteria donor	X			2.618	1.656, 4.141	**<0.001**	2.061	1.447, 2.934	**<0.001**
Cold ischemia time (h)	X			1.040	1.008, 1.074	**<0.05**	1.029	1.004, 1.053	**<0.05**
Recipient gender (fem.)	X			X			0.628	0.443, 0.891	**<0.01**
Post-transplant diabetes	X			X			0.564	0.358, 0.889	**<0.05**
HLA type I mismatch	X			X			1.188	1.022, 1.381	**<0.05**

**Table 8 jcm-14-00078-t008:** Factors influencing death-censored kidney graft survival in a 10-year follow-up period after kidney transplantation—multivariate linear regression analysis. ≥60 group—recipients aged 60 years and older (study group); <60 group—recipients younger than 60 years (control group). HR—Hazard Ratio; CI—Confidence Interval; X—non-significant.

	<60 Group			≥60 Group			Stratified		
Characteristic	HR	95% CI	*p*	HR	95% CI	*p*	HR	95% CI	*p*
Multiorgan donor	0.431	0.238, 0.781	**<0.01**	X			0.469	0.294, 0.747	**<0.001**
Any acute rejection	3.481	1.868, 6.488	**<0.001**	X			2.385	1.411, 4.031	**<0.001**
Surgical complication	2.320	1.176, 4.577	**<0.05**	2.398	1.043, 5.512	**<0.05**	2.452	1.463, 4.109	**<0.001**
Re-transplantation	2.031	1.022, 4.033	**<0.05**	X			X		
Donor age	1.050	1.015, 1.087	**<0.01**	X			X		
Extended criteria donor	X			3.844	1.762, 8.385	**<0.001**	2.512	1.552, 4.067	**<0.001**
Dialysis duration (mths)	X			1.018	1.003, 1.032	**0.017**	1.010	1.002, 1.018	**<0.05**
HLA type I mismatch	X			X			1.283	1.025, 1.608	**<0.05**

## Data Availability

The data presented in this study are available on request from the corresponding author. The data are not publicly available due to privacy.

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
