# Peer review of "Long-Term Results of Kidney Transplantation in Patients Aged 60 Years and Older"

_jcm, 2024, doi:10.3390/jcm14010078_

Round 1

Reviewer 1 Report

Comments and Suggestions for Authors

The methodology and study results are clear and well described.

Some comments about Table 4 - comparison of creatinine which could be lower in the group older than 60 years due to lower muscle mass index. Comparison of eGFR would be more precise.

In the 3rd conclusion it is mentioned that "parameters that affect the long-term results of kidney transplantation in the elderly age are: the transplantation of a kidney procured from an extended criteria donor and the duration of dialysis therapy". In the results part (lines 208 - 210) there is also importance of longer CIT so I would suggest to include this to the conclusion.

Minor grammar corrections are needed.

Reviewer 2 Report

Comments and Suggestions for Authors

Reviewer's comments:

·        I think the title of the manuscript should be: Long-Term Results of Kidney Transplantation in Patients Aged 60 Years and Older in Poland

·        what criterion were the authors guided by when they took 60 as the age limit, i.e. on the basis of which the study population was divided into two age categories under and over 60 years

·        at the end of the introductory part, the authors state the aim of the work in bold letters, highlighting two goals. It is necessary to unify the goals and state them in the form of a paragraph

·        correct the headers of Table 1, so that n, ie the number of respondents, is in the second column of the header

·        I think that table 2, which is an integral part of the methodological section of the manuscript, should not contain data related to testing the differences between the two subgroups of subjects, in the methodology it is necessary to state the characteristics of each of the examined groups, including and excluding criteria, to emphasize that the controls were matched by age and gender in relation to the first group of respondents, everything else should be an integral part of the results

·        the paragraph related to statistical methods requires correction, it is quite broad and confusingly written, concisely list the main statistical tests that were used in the analysis of the collected data

·        in the results, table 1 from the methodological section is again commented on, the same applies to table 2, exactly what I drew attention to, the tables are organized so that they are part of the results, and in the methodological part only an overview of the situation by groups of respondents and the corresponding criteria the authors managed

·        pay attention to minor mistakes in the language, eg „sereum creatinine“, etc

·        why did the authors decide to only analyze the concentration of creatinine and not the other uremic toxins; the biochemical profile of the test subject was reduced to creatinine concentration only

·        in table 6, add data for those parameters for which statistical significance has not been established, because the table displayed like this looks half-empty, the same applies to table 7 and table 8

·        give the conclusion in the form of a paragraph, separated by paragraphs, clearly emphasize the factors that proved to be statistically significant for the outcome, paragraph 2 of the conclusion

Comments on the Quality of English Language

Minor English editing.

Round 2

Reviewer 2 Report

Comments and Suggestions for Authors

I thank the authors for their detailed answers. I think the title of the manuscript should be "Long-Term Results of Kidney Transplantation in Patients Aged 60 Years and Older in Upper Silesia, Poland" without specifying the type of study.

Also, all conclusions should be presented in the form of a single paragraph, and not as items, marked numerically (1-3).
